# New Microbe Killers: Self-Assembled Silver(I) Coordination Polymers Driven by a Cagelike Aminophosphine

**DOI:** 10.3390/ma12203353

**Published:** 2019-10-15

**Authors:** Sabina W. Jaros, Matti Haukka, Magdalena Florek, M. Fátima C. Guedes da Silva, Armando J. L. Pombeiro, Alexander M. Kirillov, Piotr Smoleński

**Affiliations:** 1Faculty of Chemistry, University of Wrocław, ul. F. Joliot-Curie 14, 50-383 Wrocław, Poland; sabina.jaros@chem.uni.wroc.pl; 2Department of Chemistry, University of Jyväskulä, FIN-40014 Jyväskulä, Finland; matti.o.haukka@jyu.fi; 3Department of Pathology, Wrocław University of Environmental and Life Sciences, ul. Norwida 31, 50-375 Wrocław, Poland; magdalena.florek@upwr.edu.pl; 4Centro de Química Estrutural, Instituto Superior Técnico, Universidade de Lisboa, Av. Rovisco Pais, 1049–001 Lisbon, Portugal; pombeiro@tecnico.ulisboa.pt; 5Research Institute of Chemistry, Peoples’ Friendship University of Russia (RUDN University), 6 Miklukho-Maklaya st., 117198 Moscow, Russia

**Keywords:** silver, coordination polymers, antimicrobial materials, coordination chemistry, metal-organic frameworks (MOFs), 1,3,5-triaza-7-phospaadamantane

## Abstract

New Ag(I) coordination polymers, formulated as [Ag(*µ*-PTAH)(NO_3_)_2_]_n_ (**1**) and [Ag(*µ*-PTA)(NO_2_)]_n_ (**2**), were self-assembled as light- and air-stable microcrystalline solids and fully characterized by NMR and IR spectroscopy, electrospray ionization mass spectrometry (ESI-MS(±), elemental analysis, powder (PXRD) and single-crystal X-ray diffraction. Their crystal structures reveal resembling 1D metal-ligand chains that are driven by the 1,3,5-triaza-7-phospaadamantane (PTA) linkers and supported by terminal nitrate or nitrite ligands; these chains were classified within a **2C1** topological type. Additionally, the structure of **1** features a 1D→2D network extension through intermolecular hydrogen bonds, forming a two-dimensional hydrogen-bonded network with **fes** topology. Furthermore, both products **1** and **2** exhibit remarkable antimicrobial activity against different human pathogen bacteria (*S. aureus*, *E. coli*, *and P. aeruginosa*) and yeast (*C. albicans*), which is significantly superior to the activity of silver(I) nitrate as a reference topical antimicrobial.

## 1. Introduction

In recent years, an excessive intake of antibiotics has caused the development of human pathogen microorganisms that are multiresistant to conventional antibacterial drugs [1,2,3,4,5]. The systematic increase of such dangerous pathogen species represents a very serious threat to modern health care systems. The World Health Organization has recently released an appeal toward research on antimicrobial resistance and the development of new agents, including topical antimicrobials and derived materials [1,2,3,4,5].

One of the possible strategies to overcome the issue of multidrug-resistant bacteria is the generation of hybrid inorganic-organic molecules, such as coordination polymers (CPs), which comprise at least one antimicrobial component [1,5,6,7,8,9,10,11,12,13]. It is also important to mention other applications of coordination polymers and MOFs, such as conductive and luminescent materials, in molecular recognition, host-guest chemistry, gas sorption or supramolecular chirality [14,15,16]. Among different metal ions with a recognized antimicrobial potential, silver is particularly attractive; therefore, a body of research has been focused on antimicrobial Ag-based coordination polymers [5,6,7,8,9,10,11,12,13]. A well-established low toxicity profile, diverse structural architectures, powerful antimicrobial action, attractive argentophilic interactions, as well as the ability to slowly release Ag^+^ ions make these materials very promising candidates for antimicrobial agents. However, due to the issues of light stability and solubility in biological media, research on antimicrobial Ag(I) CPs remains a largely immature area which has great potential for the development of new, inexpensive, and highly efficient topical antimicrobials [5,6,7,8,9,10,11,12,13].

In pursuit of our general interest in this field, we developed different silver(I)-PTA based CPs with interesting antimicrobial properties (PTA = 1,3,5-triaza-7-phospahaadamante) [17,18,19,20,21,22,23]. In particular, we demonstrated that a P,N-type ligand such as PTA can be explored as a perfect multitopic building block for preparing aqua-soluble and stable silver coordination networks [17,18,19,20,21,22,23]. The unique steric and donor/acceptor properties of PTA blocks permit the creation of not only various coordination environments around silver centers but also the stabilization of metal-organic architectures, giving rise to a limited number of air- and light-stable Ag^+^-releasing antibacterial and antifungal silver-PTA networks [17,18,19,20,21,22,23,24,25,26,27,28,29,30,31,32]. The Ag-O, Ag-N, Ag-S, and Ag-P coordination environments provide a source of bioactive silver ions, which can undergo a slow release in aqueous medium and interact with the bacterial peptidoglycan cell wall, thus causing a strong antimicrobial efficiency.

Therefore, motivated by the still under-explored potential of PTA and derived Ag(I)-PTA coordination polymers, herein, we focused our attention on assembling such coordination polymers in the presence of nitrate and nitrite anions as supporting ligands. Our choice of nitrate/nitrite moieties is governed by their solubility, their ability to act as labile ligands and their common presence in physiological media, as well as a recognized application of AgNO_3_ as a potent topical antimicrobial drug and a component of antimicrobial cosmetic formulations [33,34,35,36,37].

Thus, in this study, we describe a facile self-assembly synthesis, full characterization, structural features, and the antimicrobial properties of two new 1D coordination polymers, namely [Ag(*µ*-PTAH)(NO_3_)_2_]_n_ (**1**) and [Ag(*µ*-PTA)(NO_2_)]_n_ (**2**).

## 2. Experimental

### 2.1. Materials and Methods 

Synthesis of compounds was carried out at room temperature (r.t.) in air. PTA (1,3,5-Triaza-7-phosphaadamantane) was synthesized following a reported procedure [38,39]. All other chemicals and solvents (analytical reagent grate) were used without further purification. C/N/H elemental analyses were performed by the Laboratory of Elemental Analysis at Faculty of Chemistry, University of Wrocław. Infrared (IR) spectra were measured on a Bruker IFS 1113v (Germany, Ettlingen) or BIO-RAD FTS 3000MX (BIO-RAD, Paris, France) instrument in the 4000–400 cm^−1^ range (abbreviations: vs—very strong, s—strong, m—medium, w—weak, br.—broad, sh.—shoulder). Mass spectra were recorded using a 500-MS ion trap mass spectrometer equipped with an ESI ion source (Varian, Inc., Palo Alto, CA, USA). PXRD measurements were made on a Bruker D8 ADVANCE diffractometer using Cu-Kα radiation (*λ* = 1.5418 Å) in the 2θ range of 5−60°. ^1^H and ^31^P{^1^H} NMR spectra were run in D_2_O and DMSO-*d6* solutions (r.t., ~25 °C) using a Bruker 500 AMX spectrometer. ^1^H and ^31^P{^1^H} chemical shifts are relative to Me_4_Si and external H_3_PO_4_ (85% in H_2_O), respectively.

### 2.2. Antibacterial and Antifungal Activity Studies

The antimicrobial activity of the tested compounds, as well as AgNO_2_ and AgNO_3_, used as a reference, was evaluated using the method of serial dilutions according to Grove and Randall [40,41,42]. Two reference strains were obtained from the Polish Collection of Microorganisms of the Institute of Immunology and Experimental Therapy in Wroclaw, namely *Staphylococcus aureus* PCM 2054 (=ATCC 25923) and *Escherichia coli* PCM 2057 (=ATCC 25922). Additionally, two clinical strains (*Pseudomonas aeruginosa* n=1 and *Candida albicans* n = 1) were selected. The clinical strains were isolated from a veterinary specimen and then identified as described previously [22]. In order to prepare microbial inoculum, an overnight culture of each strain was diluted (1:1000) using Antibiotic Broth (AB; medium containing [g L^−1^ (H_2_O)]: Dextrose 1.0; K_2_HPO_4_ 3.68; Beef Extract 1.5; Peptone 5.0; KH_2_PO_4_ 1.32; NaCl 3.5; Yeast Extract 1.5 [40,41,42,43]). Using 48-well plates, working dilutions of the tested substances in AB were prepared. The final concentrations of the substances ([µg mL^−1^]: 60, 50, 40, 30, 20, 10, 9, 8, 7, 6, 5, 4, 3, 2, and 1) were obtained in a well by combining 0.9 mL of the working dilution and 0.1 mL of microbial inoculum. In addition, the antimicrobial activity of PTA was examined using the same method, proving it is inactive at the maximum concentration. Broth sterility and growth controls were performed. The inoculated plates were incubated at 37 °C for 24 h. The minimum inhibitory concentration (MIC, *µ*g mL^−1^) was defined as the lowest concentration of the compound that fully inhibited the growth of bacteria or fungi. For comparison, the MIC values were normalized for the molar content of silver in **1** and **2** and are also represented in a *n*mol mL^−1^ scale.

### 2.3. Synthesis and Analytical Data

**[Ag*(µ*-PTAH)(NO_3_)_2_]_n_** (**1**). A suspension (5 mL H_2_O/10 mL MeOH) of silver oxide (0.1 mmol, 23 mg) was combined with PTA (0.2 mmol, 31.4 mg) and an aqueous NH_3_ solution (1 M, 2 mL). The obtained gray suspension was stirred at r.t. for 2 h. Next, the suspension was almost completely dissolved by a dropwise addition of aqueous HNO_3_ solution (1 M, ~4 mL until pH ≈ 4.5−5). The obtained colorless solution was filtered off and the filtrate was left to slowly evaporate in air at r.t., producing colorless X-ray quality single crystals. These crystals were collected manually and dried in air to produce **1** in 80% yield (based on Ag_2_O). S_25 °C_ (in H_2_O) ≈ 3 mg mL^−1^. Anal. Calcd. For C_6_H_13_AgN_5_PO_6_ (MW 390.1): C 18.48, N 17.96, H 3.36; Found: C 18.23, N 17.30, H 3.17. IR (KBr, cm^−1^): 2968 (s, br.), 1383 (vs, br.), 1116 (m), 1026 (s), 983 (m), 950 (s), 810 (m), 771 (m), 563 (m), 445 (w). ESI-MS(±) (H_2_O/MeOH), MS(+) *m/z*: 158 (40%) [PTAH]^+^, 421 (100%) [Ag(PTA)_2_]^+^, 484 (10%) [Ag(PTA)(PTAH)(NO_3_)]^+^, 748; MS(–) *m/z*: 231 (10%) [Ag(NO_3_)_2_]^−^. ^1^H NMR (*δ*, D_2_O): 4.80 and 4.70 (2d, J_AB_ = 14 Hz, 6H, NCH_2_N, NCH_2_N^+^, PTA); 4.34 (d, 2J_PH_ = 2.7 Hz, 6H, PCH_2_N, PCH_2_N^+^, PTA); ^1^H NMR (*δ*, DMSO-*d_6_*): 4.88 and 4.76 (2d, J_AB_ = 12 Hz, 6H, NCH_2_N, NCH_2_N^+^, PTA); 4.08 (br s, 6H, PCH_2_N, PCH_2_N^+^, PTA); ^31^P{^1^H} NMR (*δ*, D_2_O): −74.9 (br s, PTA); ^31^P{^1^H} NMR (*δ*, DMSO-d_6_): −82.8 (br s, PTA).

**[Ag(*µ*-PTA)(NO_2_)]_n_** (**2**). Solid PTA (0.2 mmol, 31.4 mg) was introduced into a solution (7 mL MeOH/3 mL H_2_O) of silver nitrite (0.2 mmol, 30.8 mg). The obtained mixture was stirred for 1 h to produce a yellowish-white solid, which was completely dissolved by adding dropwise an aqueous solution of NH_3_ (1 M, ~0.6 mL, until pH ≈ 9). The obtained yellowish solution was filtered off and a transparent filtrate was left to evaporate slowly at r.t. in air, resulting in light yellow, X-ray quality single crystals. These crystals were isolated manually and dried in air to give **2** in 60% yield (based on AgNO_2_). S_25 °C_ (in H_2_O) ≈ 3 mg mL^−1^. Anal. Calcd. for C_6_H_12_AgN_4_PO_2_ (MW 311.0): C 23.2, N 18.0, H 3.9; Found: C 23.1, N 17.80, H 3.50. IR (KBr, cm^−1^): 2935 (m) ν(H_2_O), 1438 (m), 1413 (m), 1270 (vs), 1095 (m), 1037 (m), 1014 (s), 963 (s), 949 (s), 897 (m), 807 (m), 797 (m), 749 (m), 725 (w), 596 (m), 584 (m), 563 (w), 449 (m). ESI-MS(±) (H_2_O/MeOH), MS(+) *m/z*: 158 (40%) [PTAH]^+^, 421 (100%) [Ag(PTA)_2_]^+^, 576 (10%) [Ag_2_(PTA)_2_(NO_2_)]^+^, 731 (10%) [Ag_2_(PTA)_3_(NO_2_)]^+^; MS(–) *m/z*: 198 (25%) [Ag(NO_2_)_2_]^−^. ^1^H NMR (*δ*, D_2_O): 4.3 and 4.53 (2d, J_AB_ = 12 Hz, 6H, NCH_2_N, PTA); 4.26 (br s, 6H, PCH_2_N, PTA); ^1^H NMR (*δ*, DMSO-d_6_): 4.56 and 4.40 (2d, J_AB_ = 15 Hz, 6H, NCH_2_N, PTA); 4.24 (br d, 2JP_H_ = 2.6 Hz, 6H, PCH_2_N, PTA); ^31^P{^1^H} NMR (*δ*, D_2_O): −78.8 (br s, PTA); ^31^P{^1^H} NMR (*δ*, DMSO-d_6_): −84.1 (br s, PTA).

The obtained silver compounds **1** and **2** were air stable for at least half a year in the solid state (even if exposed to light) and for several days in a mixture of DMSO-*d_6_* and D_2_O. When testing the stability in solution, the compounds were introduced into NMR tubes followed by dissolution in a mixture of 0.3 mL of DMSO-*d_6_* and 0.3 mL of D_2_O in air atmosphere. The ^31^P{^1^H} NMR spectra showed that no evident changes occurred upon keeping the dissolved samples for several days at room temperature.

### 2.4. X-Ray Crystallography

The single crystals of **1**·and **2** were mounted in an inert oil within the cold gas stream of the diffractometer (SuperNova, Single source (Mo) at offset, Eos). Structures were solved by direct methods with the SHELXS-97 program. Absorption correction was made with CrysAlisPro, Agilent Technologies (Santa Clara, CA, USA), Version 1.171.36.24. Structures were refined with SHELXL-97. H atoms were geometrically placed and constrained to ride on the corresponding parent atoms, with C–H = 0.99 Å^3^, and U_iso_ = 1.2Ueq (parent atom). In **2**, both the PTA and NO_2_^−^ ligands were disordered over two sites with equal occupancies.

Crystal data for **1**: C_6_H_13_AgN_5_O_6_P, *M* = 390.05, λ = 0.71073 Å (Mo-Kα), *T* = 170(2) K, monoclinic, space group *P*2_1_/*n*, *a* = 9.0280(4) Å, *b* = 12.6022(5) Å, *c* = 10.4447(4) Å, *α* = 90°, *β* = 91.680(4)°, *γ* = 90°, *V* = 1187.82(8) Å^3^, *Z* = 4, Dc = 2.181 g/cm^3^, *μ* = 1.866 mm^−1^, 9089 reflections collected, 3118 independent, *I* > 2*σ(I)* (*R_in_*_t_ = 0.0268), *R*_1_ = 0.0236, *wR*2 = 0.0565, GOF 1.066. CCDC 1946793.

Crystal data for **2**: C_6_H_12_AgN_4_O_2_P, *M* = 311.04, *λ* = 0.71069 Å (Mo-Kα), *T* = 150(2) K, monoclinic, space group P 21/n, *a* = 7.3771(2) Å, *b* = 9.3736(3) Å, *c* = 6.9511(2) Å, *α* = 90°, *β* = 100.107(2)°, *γ* = 90°, *V* = 473.21(2) Å^3^, *Z* = 2, *D_c_* = 2.183 g/cm^3^, *μ* = 2.278 mm^−1^, 5113 reflections collected, 911 independent, *I > 2σ(I)* (R_i*nt*_ = 0.0325), *R*_1_ = 0.0135, *wR*_2_ = 0.0317, GOF 1.054. CCDC 1946794.

## 3. Results and Discussion 

### 3.1. Synthesis and Characterization of **1** and **2**

In pursuit of our general interest in the self-assembly generation of different PTA-containing coordination compounds [17,18,19,20,21,22,23,24,25,26,27,28,29,30,31], herein, we attempted the preparation of new silver(I) coordination polymers. The 1D CP [Ag(*µ*-PTAH)(NO_3_)_2_]_n_ (**1**) was self-assembled from a reaction mixture containing an access of HNO_3_ and an alkaline solution (NH_3_ aq., MeOH/H_2_O) of silver(I) oxide and PTA at 25 °C (Scheme 1). Initially, we intended to apply a resembling synthetic procedure using silver(I) nitrate (for **1**) or silver nitrite (for **2**), PTA, and an excess of HNO_3_ along with aqueous NH_3_. However, this synthetic pathway resulted in the formation of **1** in all cases. Thereby, compound [Ag(*µ*-PTA)(NO_2_)]_n_ (**2**) was obtained by treating silver(I) nitrite at 25 °C in H_2_O/MeOH solvent with PTA, followed by the alkalization with aqueous NH_3_ (Scheme 1).

Both Ag(I) CPs precipitated as microcrystalline solids and were isolated as stable and yellowish-white materials. The structures of **1** and **2** were determined by single crystal X-ray diffraction and the product formulae were further confirmed by IR, ^1^H and ^31^P{H} NMR spectroscopies, PXRD (Appendix A), elemental analysis and electrospray ionization mass spectrometry, ESI(±)-MS.

The IR spectra of **1** and **2** shows a characteristic set of vibrations in the ranges 2968–2935 and 1440–400 cm^−1^, corresponding to coordinated [PTAH]^+^ or PTA ligands, which are shifted to higher wavelengths by 20−30 cm^−1^ when compared to uncoordinated aminophosphine blocks. There are also the typical stretching vibrations of coordinated nitrate [1383 cm^−1^] and nitrite [1270 cm^−1^] ligands. The ^1^H NMR spectra of **1** and **2** in D_2_O and DMSO-*d_6_* show typical signals owing to the coordinated PTA and PTAH moieties, respectively. The ^31^P NMR resonances of PTA appear at a lower field when compared with those of uncoordinated PTA, thus proving its coordination to Ag center [18]. The ESI-MS plots of both compounds are dominated by the same silver(I) cationic products [Ag(PTA)_2_]^+^ (*m/z* 421). Other fragments were detected at *m/z* 484, 576, and 731, and identified as [Ag(PTA)(PTAH)(NO_3_)]^+^ (**1**), [Ag_2_(PTA)_2_(NO_2_)]^+^ (**2**), and [Ag_2_(PTA)_3_(NO_2_)]^+^ (**2**), respectively. Moreover, the ESI-MS data match the theoretical, isotopic distribution of silver containing species, and agree with previously reported fragmentation patterns for Ag-PTA CPs [17,18,19,20,21,22,24].

### 3.2. Crystal Structures of **1** and **2**

A single-crystal X-ray structural analysis shows that compound **1** is a 1D coordination polymer that is driven by µ-PTAH linkers (Figure 1). An asymmetric unit is composed of Ag1 atom, one µ-PTAH block, and two terminal nitrate ligands (Figure 1a). The Ag1 center is four-coordinate and features a distorted {AgPNO_2_} geometry that is taken by the N1 and P1 donors [Ag1−P1 2.3590(5), Ag1 N1 2.4742(17) Å] from two different *µ*-PTAH moieties and two nitrate O1 and O4 atoms [Ag1−O1 2.3452(13), Ag1−O4 2.5982(14) Å]. The *τ*_4_ parameter calculated for the Ag1 atom (*τ*_4_
*~*0.76) indicates a geometry between the trigonal pyramid and seesaw. 

The bond lengths around the Ag1 center are comparable to literature data for related Ag(I) coordination networks [1]. The protonated [PTAH]^+^ block displays a bidentate P, N-coordination mode and behaves as a µ-linker between the adjacent Ag1 centers. As a result, the ~Ag−PTAH−Ag−PTAH~ metal-ligand chains are generated (Figure 1b) with the Ag1···Ag1 separation of 6.802 Å. From a topological perspective, such chains are described as a uninodal two-connected net of the **2C1** topological type (Figure 1c). The neighboring chains are held together by intermolecular bifurcated N2−H2···O5(O6) hydrogen bonds between the NH functionality of [PTAH]^+^ and two O acceptors of the nitrate ligand. Such H-bonding interactions result in the 1D→2D extension of the structure and lead to a generation of 2D hydrogen-bonded layers. After simplification, such layers can be topologically described as a uninodal 3-connected net with the **fes** [Shubnikov plane net (4.8^2^)] topology and point symbol (4.8^2^).

Similarly to **1**, the structure of **2** bears the 1D metal-ligand chains that are constructed from the [Ag(NO_2_)] nodes and the µ-PTA linkers (Figure 2). The main distinctive features of **2** vs. **1** concern the presence of only one terminal ligand (bidentate NO_2_^−^ moiety) and a neutral non-protonated µ-PTA block. Four-coordinate Ag1 atoms adopt a distorted {AgPNO_2_} environment that is better described as a seesaw geometry (*τ*_4_ ~0.65). The Ag1 center is coordinated by the P1 and N2 atoms [Ag1−P1 2.349(3), Ag1−N2 2.399(9) Å] coming from two distinct µ-PTA blocks and a pair of symmetry generated O1 atoms [Ag1−O1 2.414(3) Å] from the terminal bidentate NO_2_^−^ ligand. From a topological viewpoint, the 1D metal-ligand chains in **2** are similar to those of **1** (**2C1** topology, Figure 1c).

### 3.3. Antimicrobial Activity of **1** and **2**

The antimicrobial properties of metallic silver and its compounds have been known for centuries. However, aside from topical silver nitrate and silver sulfadiazine antimicrobials, the use of silver compounds in antimicrobial treatment has not been common due to the widespread distribution of antibiotics. Given the increasing growth of problems due to multiresistant bacteria, the development of novel topical antimicrobials and formulations has been on the agenda of the World Health Organization [27,28]. 

As a continuation of our efforts toward the search for new antimicrobial silver(I)-based agents, herein, we studied the antimicrobial potential of **1** and **2** against four model human pathogens that included Gram-positive and Gram-negative bacteria and yeast. Hence, compounds **1** and **2** were evaluated against *P. aeruginosa, E. coli*, *S. aureus*, and *C. albicans*, and the minimal inhibitory concentrations (MIC, µg·mL^−1^) were obtained by a serial dilution method (Table 1). 

The MIC values were also recalculated for the molar content of silver ion (normalized MIC, nmol·mL^−1^) and compared with the antimicrobial activity of a reference antimicrobial agent (AgNO_3_) as well as silver(I) nitrite.

The screening of **1** and **2** shows their different antibacterial activity against Gram-negative and Gram-positive bacteria. Generally, both compounds reveal a resembling trend to strongly inhibit the growth of *E. coli* and *P. aeruginosa* pathogens with the MIC and normalized MIC values in the rage of 3−6 µg·mL^−1^ and 10−15 nmol·mL^−1^, respectively. Comparison with the AgNO_3_ reference indicates that both CPs exhibit a 2 to 5-fold superior antibacterial activity against Gram-negative bacteria.

In the case of Gram-positive bacteria, a remarkably higher antibacterial activity was observed against the *S. aureus* strain, for which the lowest normalized MIC value of 19 nmol·mL^−1^ was attained for **2**. Hence, this silver derivative is two to six times more active in antimicrobial efficiency if compared with **1** and AgNO_3_. It should be highlighted that the antimicrobial activity of compound **2** in relation to other reported silver coordination polymers is relatively high, both against Gram-negative and Gram-positive bacteria [44,45,46].

The distinct antimicrobial behavior between **1** and **2** against the Gram-positive and Gram-negative bacteria is probably related to the nature and structure of these bacteria. In fact, thicker cell walls of Gram-positive bacteria, together with bigger steric hindrance of compound **1,** most probably reduce the penetration rate of silver ions into the cell, thus explaining a suppressed antimicrobial activity against *S. aureus*. Moreover, ESI-MS measurements indicate a slightly different speciation of compounds **1** and **2** in solution. Despite some similarities in speciation, we believe that the presence of different cationic species, namely [Ag(PTA)(PTAH)(NO_3_)]^+^ (in **1**) and [Ag_2_(PTA)_2_(NO_2_)]^+^ (in **2**), may explain the distinct antimicrobial activity of these compounds. The extension of this study toward a detailed investigation of the mechanistic aspects of the observed antimicrobial activity, as well as the application of the obtained compounds as components of antimicrobial composite materials, is currently in progress and will be reported elsewhere.

## 4. Conclusions

By employing the self-assembly approach and using 1,3,5-triaza-7-phospahaadamantane as a main mulitopic building block, we generated, isolated, and fully characterized two new Ag(I) CPs [Ag(*µ*-PTAH)(NO_3_)_2_]_n_ (**1**) and [Ag(*µ*-PTA)(NO_2_)]_n_ (**2**). Both compounds extend a still limited family of Ag-based metal-organic architectures assembled from a versatile, water-soluble, and cagelike aminophosphine (PTA). The obtained products also display remarkable antibacterial and antifungal activity against four model pathogens. In fact, compounds **1** and **2** are particularly efficient at inhibiting the growth of *E. coli* and *P. aeruginosa*, showing normalized MIC values that are significantly lower than those of the reference AgNO_3_ antibacterial. The differences observed in the antimicrobial behavior of **1** and **2** might be related to the type of anionic terminal ligand (nitrate vs. nitrite), protonated or neutral character of [PTAH]^+^/PTA moieties, as well as the stability of compounds in Antibiotic Broth medium. Notably,, when compared with other evaluated eAg-based coordination polymers evaluated to date, compound **2** exhibits a superior antibacterial activity.

In summary, we believe that the self-assembly procedure applied in the present study should be extended to other types of simple anionic ligands and additional building blocks aiming at the generation of stable, tailorable, and bioactive Ag-PTA based coordination networks with even more pronounced antimicrobial action. Furthermore, the results obtained may open new horizons in the design of topical Ag(I)-containing antimicrobial materials and formulations.

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
