# Peer review of "New Microbe Killers: Self-Assembled Silver(I) Coordination Polymers Driven by a Cagelike Aminophosphine"

_materials, 2019, doi:10.3390/ma12203353_

Round 1

Reviewer 1 Report

The authors report on two silver(I) coordination polymers self-assembled as crystalline solids. The products were characterized by IR and NMR spectroscopy, ESI-MS(±), elemental analysis, powder and X-ray single-crystal diffraction.

The recognized antimicrobial activity of silver ion is well focused and analysed, and discussion about single crystal X-ray diffraction analysis well described.

MIC antimicrobial activity and structural features are exhaustively presented, but Introduction must be implemented keeping in mind that one of the main topics, as the title says, is the synthesis of CPs. This reach varied class of compounds, both crystalline and amorphous, are not described by the authors, nor recent inherent literature cited, as for example: Diana, R., Panunzi, B., Shikler, R., Nabha, S., Caruso, U., “Highly efficient dicyano-phenylenevinylene fluorophore as polymer dopant or zinc-driven self-assembling building block”  Inorganic Chemistry Communications, 2019 (104), pp.  145-149;   Panunzi, B., Concilio, S., Diana, R., Shikler, R., Nabha, S., Piotto, S., Sessa, L., Tuzi, A., Caruso, U. “Photophysical Properties of Luminescent Zinc(II)‒Pyridinyloxadiazole Complexes and their Glassy Self-Assembly Networks” European Journal of Inorganic Chemistry, 2018 (23), pp. 2709-2716.

In line 26 and in other cases the author use “metal-organic” but it is incorrect: the term is used properly only in the case of a C-M bond.

In line 24: air and light-stable crystalline solids. This concept is emphasized again in the text. What kind of quantitative stability test has been performed?

 1H and 31P{1H} NMR spectra of compounds were recorded. Please, add also 13C NMR spectra as commonly reported.

In Results and Discussion lines 252 and following:  Comparison with the AgNO3 reference indicates that both CPs exhibit a 2- to 5-fold superior antibacterial activity…..and line 256: Hence, this silver derivative is 2 to 6 times more active in antimicrobial efficiency if compared with 1 and AgNO3. And also, in Conclusions, lines 278-279: Notably, if compared with other evaluated so far Ag-based coordination polymers, compound 2 exhibits superior antibacterial activity. Please, add appropriate references and effective comparison data to help the reader.

English is correct, some typos must be emended. At this stage the paper can be considered for publication in Materials after the recommended revisions.

Author Response

We thank the Reviewer for a positive evaluation of our work and for very valuable suggestions aiming at its further improvement. The manuscript has been revised in light of those suggestions and the necessary amendments/corrections have been introduced.

Reviewer 2 Report

The authors have prepared two silver coordination polymers with antimicrobial behavior.  Both structures appear to be new.  There are a large number of Ag coordination polymers in the literature. The work is quite routine and contribute less chemistry and materials properties.

I do not see discussion of the results received in this investigation. The authors should give the design chemistry of both new coordination polymers that relative to the antimicrobial applications. Further details and mechanism also need to show. 

In this case, the authors should better specify in discussion and in conclusions, what are the differences between already known and new compounds and what structural or functional features make the novelty of these coordination polymers.

I do not recommend publication of current status of this manuscript in Materials.

Author Response

We thank the Reviewer for evaluation of our work and for very valuable suggestions aiming at its further improvement.

Round 2

Reviewer 2 Report

The authors did well revision jobs. This manuscript can be accepted for publication at current status.